# Lymphoid origin of intrinsically activated plasmacytoid dendritic cells in mice

Alessandra Machado Araujo[1†], Joseph D Dekker[1*†], Kendra Garrison[1], Zhe Su[2], Catherine Rhee[3], Zicheng Hu[3], Bum-Kyu Lee[3], Daniel Osorio[2], Jiwon Lee[1], Vishwanath R Iyer[3], Lauren IR Ehrlich[3], George Georgiou[3], Gregory Ippolito[3], Stephen Yi[2*], Haley O Tucker[3*]

[1]Department of Chemical Engineering, The University of Texas at Austin, Austin, United States; [2]Department of Biomedical Engineering, and Livestrong Cancer Institutes, The University of Texas at Austin, Austin, United States; [3]Department of Molecular Biosciences, The University of Texas at Austin, Austin, United States

**\*For correspondence:**
joedekker@gmail.com (JDD);
stephen.yi@austin.utexas.edu
(SY);
haleytucker@austin.utexas.edu
(HOT)

[†]These authors contributed
equally to this work

**Reviewing Editor:** Dipyaman
Ganguly, Indian Institute of
Chemical Biology, India

**Abstract** We identified a novel mouse plasmacytoid dendritic cell (pDC) lineage derived from the common lymphoid progenitors (CLPs) that is dependent on expression of *Bcl11a*. These CLP-derived pDCs, which we refer to as 'B-pDCs', have a unique gene expression profile that includes hallmark B cell genes, normally not expressed in conventional pDCs. Despite expressing most classical pDC markers such as SIGLEC-H and PDCA1, B-pDCs lack IFN-α secretion, exhibiting a distinct inflammatory profile. Functionally, B-pDCs induce T cell proliferation more robustly than canonical pDCs following Toll-like receptor 9 (TLR9) engagement. B-pDCs, along with another homogeneous subpopulation of myeloid-derived pDCs, display elevated levels of the cell surface receptor tyrosine kinase AXL, mirroring human AXL[+] transitional DCs in function and transcriptional profile. Murine B-pDCs therefore represent a phenotypically and functionally distinct CLP-derived DC lineage specialized in T cell activation and previously not described in mice.

## eLife assessment

This **valuable** study reports that while most plasmacytoid dendritic cells (pDCs) originate from common dendritic cell precursors, approximately 20% are derived from lymphoid progenitors shared with B cells. The methodology used and the evidence are **solid**, and further demonstrate the distinct transcription factor requirements and activities of this subset of pDCs, although the functional significance of this dendritic cell subset will require further elucidation. The findings will be of great interest for those interested in the developmental and functional biology of the immune system.

## Introduction

Plasmacytoid dendritic cells (pDCs) specialize in the production of type I interferons (IFNs) and promote antiviral immune responses following engagement of pattern recognition receptors. They have been mainly implicated in the pathogenesis of autoimmune diseases that are characterized by a type I IFN signature (notably, IFN-α). Yet, pDCs have also been shown to be able to induce tolerogenic immune responses (*Di Domizio and Cao, 2013*; *Swiecki and Colonna, 2015*; *Sisirak et al., 2014*; *Rönnblom and Pascual, 2008*). Due to the clinical significance of pDCs, several studies aimed at mapping pDC lineage derivation have emerged recently. Yet, a complete understanding of pDC development and their existing subsets in mice is still lacking. The transcription factor 4 (TCF4) is required for pDC development and for lineage identity (*Cisse et al., 2008*; *Ghosh et al., 2010*; *Reizis et al., 2011*). TCF4 is a component of a multiprotein complex that includes both positive and negative regulators

(*Swiecki and Colonna, 2015*; *Grajkowska et al., 2017*). One of these components, the transcription factor BCL11a, which is also essential for B cell development (*Liu et al., 2003*; *Ippolito et al., 2014*; *Yu et al., 2012*), induces *Tcf4* transcription and initiates a positive feedback loop with TCF4 to maintain pDC lineage commitment and function (*Ippolito et al., 2014*; *Yu et al., 2012*; *Wu et al., 2013*).

Unlike their conventional dendritic cell (cDC) counterparts, pDCs express transcriptional regulators and markers associated with B-lymphocyte development in addition to BCL11a (e.g. B220, SPIB) (*Pelayo et al., 2005*; *Sathe et al., 2013*). These features, along with the established generation of pDCs from myeloid restricted precursors (*Feng et al., 2022*; *Harman et al., 2006*), have made it difficult to define pDC lineage affiliation (*Reizis et al., 2011*; *Pelayo et al., 2005*; *Shigematsu et al., 2004*; *Wang and Liu, 2004*). This has led to the hypothesis that pDC subsets may have distinct origins derived from either the common lymphoid progenitor (CLP) or the common myeloid progenitor (CMP) (*Sathe et al., 2013*; *Nikolic et al., 2002*). Beyond the complex nature of their lineage, pDCs with different functional attributes (e.g. variable IFN-α expression levels) or different surface markers (e.g. CD19$^+$ pDCs detected in tumor-draining lymph nodes) have also been identified (*Pelayo et al., 2005*; *Yang et al., 2005*; *Mellor et al., 2005*; *Munn et al., 2004*).

On par with the ever-increasing heterogeneity described within pDC populations, a novel AXL$^+$ dendritic cell (DC) population with many pDC-like properties has recently been discovered in human blood (*Villani et al., 2017*; *Alcántara-Hernández et al., 2017*; *Matsui et al., 2009*). While AXL$^+$ DCs express many canonical pDC protein markers (e.g. CD123, BDCA2/CD303), they also express CD2, the Ig-like lectins SIGLEC1 and SIGLEC6, as well as the activation marker CD81 (*Villani et al., 2017*). In a separate study (*Zhang et al., 2017*), a similar population of pDCs was shown to express high levels of CD5 and CD81 in mice and humans, two glycoproteins normally associated with the B cell receptor (BCR) signaling complex. The origin of AXL$^+$ transitional DCs is currently unclear and the presence of a homologous population in mice remains muddled. Here, we report the identification of a lymphoid-derived pDC subset (B-pDCs) that shares inflammatory features with myeloid-derived Axl$^+$ pDC populations in mice. We also demonstrate an in vivo requirement for *Bcl11a* in the transcriptional specification of B-pDCs.

## Results

We previously demonstrated that conditional deletion of *Bcl11a* in the hematopoietic stem cell compartment mediated by *Vav1*-Cre or by inducible *Mx1*-Cre recombinase results in complete abolishment of pDC development (*Ippolito et al., 2014*). Spurred by previous speculation of pDC origin from the CLP (*Pelayo et al., 2005*; *Wang and Liu, 2004*), we next selectively deleted floxed ($^F$) *Bcl11a* alleles in *Cd79a$^+$* cells (*Hobeika et al., 2006*; *Sakaguchi et al., 1988*), as mediated by *Cd79a*-Cre in vivo. Expression of the *Cd79a* gene (*Cd79a*) initiates at the LY6D$^+$ CLP stage, in B-cell-biased lymphoid progenitors (BLPs) (*Inlay et al., 2009*), downstream of LY6D$^-$ CLPs (*Figure 1—figure supplement 1*). *Bcl11a$^{F/F}$Cd79a-Cre* mice (cKO) and littermate controls were examined for pDC frequencies among nucleated cells in the bone marrow (BM). B220$^+$ PDCA1$^+$ pDCs (all found to be CD11c$^{int}$) were consistently and significantly reduced by an average of ~25% (24.8 ± 2.4%) in cKO mice relative to littermates (*Figure 1A and B*). Loss of B cells (B220$^{hi}$ PDCA1$^-$) served as a gauge of *Cd79a*-Cre deletion efficiency (*Figure 1A and B*). Taken together, these data indicate that a significant proportion of pDCs are derived from BLPs or BLP-derived cells and are *Bcl11a* dependent.

To expand these in vivo observations, and to confirm that this defect is intrinsic to hematological progenitor cells, we transferred BM from either BCL11A-sufficient reporter control mice (*Cd79a-Cre-YFP*) or BCL11A-deficient cKO mice (*Bcl11a$^{F/F}$Cd79a-Cre-YFP*) into lethally irradiated wildtype C57BL/6J recipients. After 8 weeks, <10% of B cells (B220$^+$ PDCA1$^-$) in the spleens of *Cd79a-Cre-YFP* recipients were YFP$^-$, confirming elimination of recipient hematopoiesis (*Figure 1D and E*). As expected, *Bcl11a$^{F/F}$Cd79a-Cre-YFP* BM resulted in significantly reduced B cell and pDC cellularity compared to *Cd79a-Cre-YFP* controls, whereas BCL11A-sufficient (YFP$^-$) pDCs persisted (*Figure 1C, E, and F*).

Approximately one-third of pDCs in the spleen of wildtype chimeras were YFP$^+$ compared to only one-fifth in BM (*Figure 1G*). This increased fraction of YFP$^+$ pDCs in the spleen suggests that BLP-derived pDCs preferentially home to that organ. While the residual YFP$^+$ pDCs in the BM might be an indication that *Bcl11a* is not shut off at the most precise developmentally critical point within the B cell lineage for complete *Cd79a$^+$* pDC ablation, other hematopoietic lineages were capable of

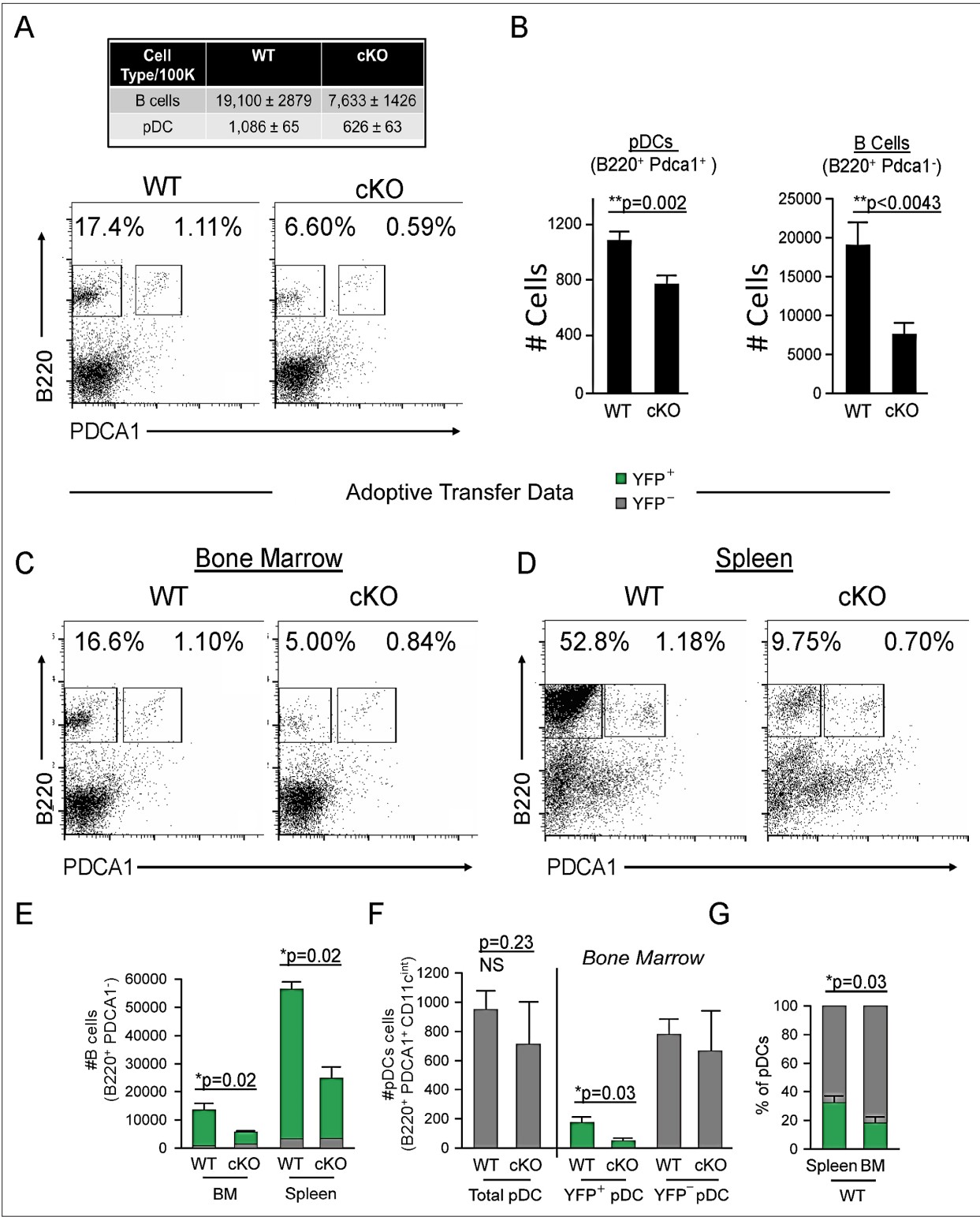

| Cell Type/100K | WT | cKO |
|---|---|---|
| B cells | 19,100 ± 2879 | 7,633 ± 1426 |
| pDC | 1,086 ± 65 | 626 ± 63 |

**Figure 1.** *Cd79a*-Cre deletion of *Bcl11a* identifies a common lymphoid progenitor (CLP)-derived subset of plasmacytoid dendritic cells (pDCs). (**A**) Representative FACS plots of pDC (gated as B220+ PDCA1+) and B cell (gated as B220+ PDCA1-) percentages in the bone marrow (BM) of *Bcl11a*^F/F *Cd79a*-Cre mice (cKO) and littermate controls. (**B**) Quantification of pDC and B cell populations in the BM of *Bcl11a*^F/F *Cd79a*-Cre mice (cKO) and littermate controls as cells/100,000 cells. (**C–D**) Flow cytometric analysis of BM and spleens of recipient mice 8 weeks post BM transplantation. BM was transferred from either BCL11A-sufficient reporter control mice (*Cd79a*-Cre-YFP) or BCL11A-deficient cKO mice (*Bcl11a*^F/F *Cd79a*-Cre-YFP) into lethally irradiated C57BL/6J recipients. (**E**) B cell numbers after BM transplantation in BM and spleens of recipient mice. (**F**) pDC numbers after BM

*Figure 1 continued on next page*

Figure 1 continued

transplantation in BM and spleens of recipient mice. (**G**) Comparison of YFP⁺ pDC percentages in the spleen and BM of recipient mice post BM transplantation. Mann-Whitney t-tests were used for all statistical comparisons. Error bars=mean±s.d. The results are representative of two experiments each containing 3–4 mice per group.

The online version of this article includes the following figure supplement(s) for figure 1:

**Figure supplement 1.** Progenitor population analysis for *Cd79a*-cre driven YFP expression.

**Figure supplement 2.** Cell distributions in the spleen of adoptively transferred recipient mice.

development in normal numbers and contained a paucity of YFP⁺ cells (*Figure 1—figure supplement 2*). Of note, BCL11A-deficient progenitors yielded higher splenic T cell chimerism at the expense of B cells and pDCs, but less than 2% of T cells were YFP⁺ (*Figure 1—figure supplement 2*, and not shown). This indicated that *Cd79a-cre* expression occurs subsequent to T-B lineage divergence, consonant with the previous observations (*Inlay et al., 2009*; *Nagaharu et al., 2022*; *Herman and Grün, 2018*) and our *Cd79a-cre* progenitor analysis in which YFP⁺ cells are confined to the BLP-derived compartment (*Figure 1—figure supplement 1*). Because of their exclusive lymphoid derivation post T-B bifurcation, we were prompted hereafter to refer to this pDC lineage as 'B-pDC'.

Relative to *Cd79a⁻* pDCs, resting B-pDCs express higher levels of MHC Class II (*Figure 2A and B*), suggesting that they may be primed for immediate response to pro-inflammatory signals. To test this hypothesis, we delivered TLR9 ligand (CpG:ODN) into *Cd79a-Cre-YFP* mice via tail vein injection, and splenic pDCs were phenotyped via flow cytometry 24 hr later. While both pDC and B-pDC compartments expanded relative to controls (*Figure 2C and D*), the YFP⁺ B-pDC fraction increased almost twofold above that of the YFP⁻ pDC fraction in numbers (*Figure 2E*). Additionally, fold increase of B-pDCs coexpressing CD83 and CD86 upon activation was also significantly higher than that of YFP⁻ pDCs (*Figure 2F and G*).These results suggested that relative to YFP⁻ pDCs, B-pDCs are intrinsically activated and primed for rapid expansion upon TLR9 engagement. To confirm the functional phenotype of B-pDCs, we tested their ability to secrete cytokines known to be elicited by pDCs after Toll-like receptor (TLR) engagement. Specifically, we tested each pDC lineage for the production of IFN-α or IL-12p40 when activated by TLR9-bound CpG oligonucleotides. We sorted B-pDCs and pDCs, engaged TLR9 with CpG:ODNs for 24 hr and collected supernatant for cytokine-specific ELISAs. IFN-α production was almost negligible in B-pDC (*Figure 2H*), yet IL-12p40 production was significantly augmented over pDC (*Figure 2H*). To test their ability to expand T lymphocytes in culture, sorted B-pDCs and pDCs were incubated with CpG:ODNs and co-cultured with freshly isolated CFSE-labeled lymphocytes. After 6 days, co-cultures were stained for CD3 and CFSE-negative cell percentages were recorded (*Figure 2I and J*). Altogether, our results showed B-pDCs were significantly better at expanding T cells in co-cultures than conventional pDCs (*Figure 2I and J*) upon TLR9 activation.

To further elucidate their phenotype at the genetic level, we performed RNA-seq analyses of purified B-pDCs and compared them to classical, myeloid-derived bulk pDCs. While the overall gene expression patterns were highly similar across the two subsets, ~1% of transcripts (~220 genes) differed significantly (q-value<0.05) (*Figure 2—figure supplement 1*). Among the top overexpressed genes in B-pDCs were *Lyz1*, *Ccr3*, *Cd86*, *Id2*, *Axl*, *Siglec1*, and *Cd81* (*Figure 2—figure supplement 1*). Differentially expressed transcripts generated Gene Ontology (GO) (*Huang et al., 2009*) or Panther (*Thomas et al., 2003*) terms including 'immune response', 'inflammatory response', 'cell activation', and 'regulation of immune response' (p=1.36 × 10⁻¹⁷, 2.15×10⁻¹², 3.67×10⁻¹¹, and 1.45×10⁻¹⁰), respectively (*Figure 2—figure supplement 1*). Gene set enrichment analysis (GSEA) revealed elevation of each of these same GO/Panther terms within the B-pDC subset as compared to a pDC-related GSEA control dataset that showed no enrichment. Next, we compared all genes expressed by both pDC populations to one another and to published (*Liu et al., 2014*) RNA-seq of BM-derived mouse pre-B cells (B220⁺ IgM⁻ Kit⁻ CD25⁺)—a post-CLP B cell progenitor (pre-B) population. As shown in *Figure 2D*, pDCs and B-pDCs expression levels were strongly correlated with one another relative to early B cells (R² values = 0.8959, 0.4145, and 0.404, respectively) (*Figure 2—figure supplement 1*). Collectively, these data support our contention that B-pDCs are functionally distinct from classical pDCs and specialize in inflammatory responses, antigen presentation, and T cell activation.

AXL⁺ transitional DCs (tDCs or AS-DCs) *Villani et al., 2017*; *Leylek et al., 2019* have been identified in humans by the high expression levels of the genes *AXL*, *CD81*, *CD86*, *LYZ2*, *C1QA*, *CD2*, and

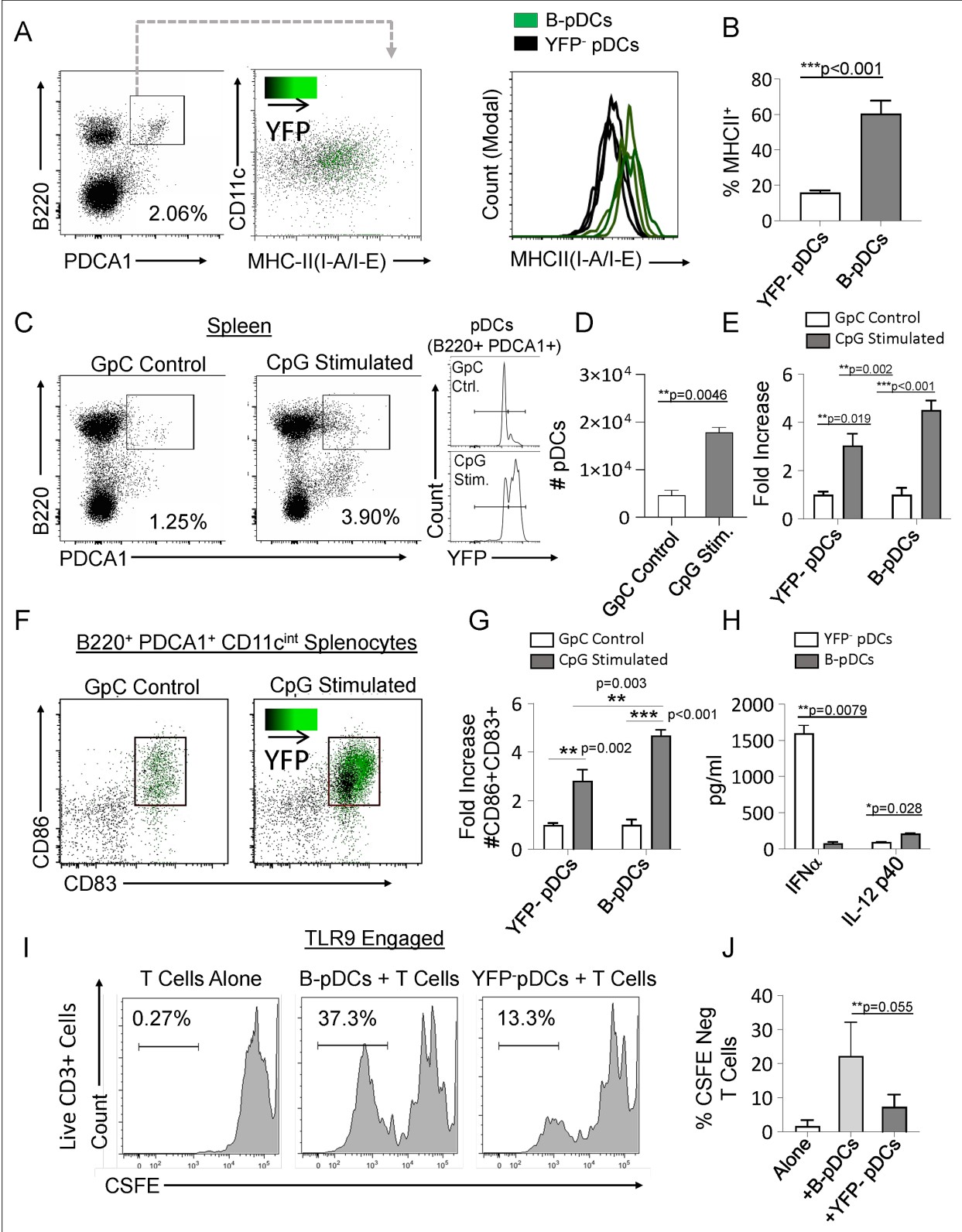

**Figure 2.** Common lymphoid progenitor (CLP)-derived B-plasmacytoid dendritic cell (B-pDC) are licensed for T cell activation. (A–B) Comparison of MHCII MFI in YFP $^+$ and YFP$^-$ pDCs from *Cd79a*-Cre-YFP reporter mice via flow cytometric analysis. (C–D) *Cd79a*-Cre-YFP mice were injected with 50 μg/mL (100 μL) of CpG:ODN or control GpC:ODN and analyzed via flow cytometry for splenic pDC numbers/100,000 cells. (E) Flow cytometry quantification of YFP$^-$ pDC and B-pDCs (YFP$^+$) fold change in cell numbers upon CpG:ODN in vivo challenge. (F–G) Fold change in CD86$^+$CD89$^+$ cell numbers for

*Figure 2 continued on next page*

*Figure 2 continued*

each pDC population in mice restimulated with CpG:ODN. Difference in CD86⁺CD89⁺ cell numbers between stimulated B-pDCs and YFP⁻ pDCs was also significant. (**H**) In vitro Toll-like receptor 9 (TLR9) engagement of B-pDC or pDC for ELISA against IFN-α orIL-12p40. (**I**) T cells were magnetically isolated from wildtype C57BL/6J mouse splenocytes using MACs columns, labeled with CSFE, and then cultured (2.5×10⁴/well) alone or with CpG:ODN activated B-pDCs or pDCs (5×10³/well) for 6 days. (**J**) The percentage of CFSE-negative CD3⁺ T cells in co-cultures were significantly higher in B-pDC compared to pDCs. Mann-Whitney t-tests were used for all statistical comparisons. Error bars = mean ± s.d. The results are representative of three independent experiments each containing at least 4 mice per group.

The online version of this article includes the following figure supplement(s) for figure 2:

**Figure supplement 1.** Transcriptional analysis identifies two populations of plasmacytoid dendritic cell (pDC) in mice: myeloid-derived classical pDC and common lymphoid progenitor (CLP)-derived B-pDC.

**Figure supplement 2.** Imaging flow cytometry as a tool to quantify AXL expression in plasmacytoid dendritic cell (pDC) populations.

*C1QB* relative to classical pDCs (*Villani et al., 2017*; *Zhang et al., 2017*; *Alcántara-Hernández et al., 2017*; *Matsui et al., 2009*). They also exhibit many classical pDC features, having previously been described phenotypically as a 'continuum' between pDC and cDC2 populations (*Zhang et al., 2017*). In order to confirm AXL⁺ pDCs are also present in mice and corroborate our RNA-seq findings, we phenotyped B-pDCs through imaging flow cytometry. First, we again verified that our primary gating scheme effectively encompassed bona fide pDCs using imaging flow cytometry. As expected, B220⁺ PDCA1⁻ cells were >95% CD19⁺ SIGLEC-H⁻ (B cells). Most importantly, >95% of B220⁺ PDCA1 ʰⁱ cells were SIGLEC-H ⁺and CD19⁻, ruling out any pDC population contamination (*Figure 2—figure supplement 2*). At the protein level, we were able to stain mouse BM cells with anti-AXL antibodies without issues and confirmed that while ~60% of B-pDCs were positive for AXL, only ~10% of YFP negative pDCs expressed this protein (*Figure 2—figure supplement 2*). Finally, imaging of BM cells showed that B-pDCs resembled classical secretory pDCs morphologically, presenting a large nucleus and defined plasma-like features, in contrast to B cells (*Figure 2—figure supplement 2*). This preliminary phenotypic analysis led us to speculate that similarly to human AXL⁺ transitional DCs, B-pDC might also prioritize their secretory capacity for copious transcription and secretion of alternative immune system modulators. Thus, we performed high-resolution 10× single-cell RNA-seq analyses of magnetically sorted PDCA1⁺ BM cells (96% purity as determined by flow cytometry) and compared the B-pDC transcriptional profile to that of B cells and other pDCs. Principal component-based clustering and UMAP visualization of sorted PDCA⁺ cells yielded 22 discrete clusters. To identify cell populations in an unbiased manner, we uploaded the top 1000 differentially expressed genes (DEGs) from each cluster into CIPR (Cluster Identity Predictor) (*Ekiz et al., 2020*), which uses the reference database ImmGen (mouse) to calculate cell identity scores (*Supplementary file 1*). Seurat clusters were named according to the consensus of the top 5 cell ID hits generated by CIPR (*Figure 3A*). We confirmed CIPR assignments by evaluating cluster DEGs and expression of markers associated with specific myeloid cell populations (*Figure 3—figure supplement 1*). PDCA1⁺ cells comprised nine distinct pDC populations (including B-pDCs, which were unanimously classified as pDCs by CIPR identity scores despite its unique *Cd79a* expression). Of note, pDC8 had one of the lowest ID scores among consensus pDCs (*Supplementary file 1*) and yielded CIPR identity hits for both pDCs and 'CMPs' within its top 5 ID hits (not shown). This suggested that this single cluster (1.7% of total cells) represented a mixed population of pDCs and differentiating CMPs expressing high levels of *Hbb* family genes (hemoglobin associated genes) under the chosen default Seurat clustering resolution settings. Because of pDC8 mixed identity, we excluded this cluster from our further single-cell transcriptional analysis. Besides pDCs, a small B cell cluster (2.75% of all cells analyzed) expressing low levels of *Bst2* (PDCA1 gene), as well as distinct subpopulations of macrophages (Mac1, -2, and -3), monocytes (Mono1, -2, and -3), and granulocytes (Granul1, -2, and -3) were also identified. Two stem-progenitor populations (Prog1 and -2) were present in our cell pool (1.12% and 2.86% of cells analyzed, respectively) (*Figure 3A and B*). The Prog2 population expressed high levels of genes associated with pDCs, including *Siglech*, *Bst2*, *Flt3*, *TCF4*, and *Irf8* and had a 'CDC' (common dendritic cell progenitor) signature according to its CIPR reference ID (*Supplementary file 2*). In addition, the majority of cells in the Prog2 cluster were *Csf1r⁻*, *Il7r⁺*, *Siglech⁺*, *Ly6d⁺* (*Figure 3—figure supplement 1*, *Supplementary file 2*), resembling the pro-pDC myeloid precursor population described by Rodriguez et al., as well as Dress and colleagues (*Rodrigues et al., 2018*; *Dress et al., 2019*). In contrast, Prog1, which clustered near B cells and was identified as 'MLP' (multi-lymphoid progenitors entering the CLP stage), expressed several pre/pro B

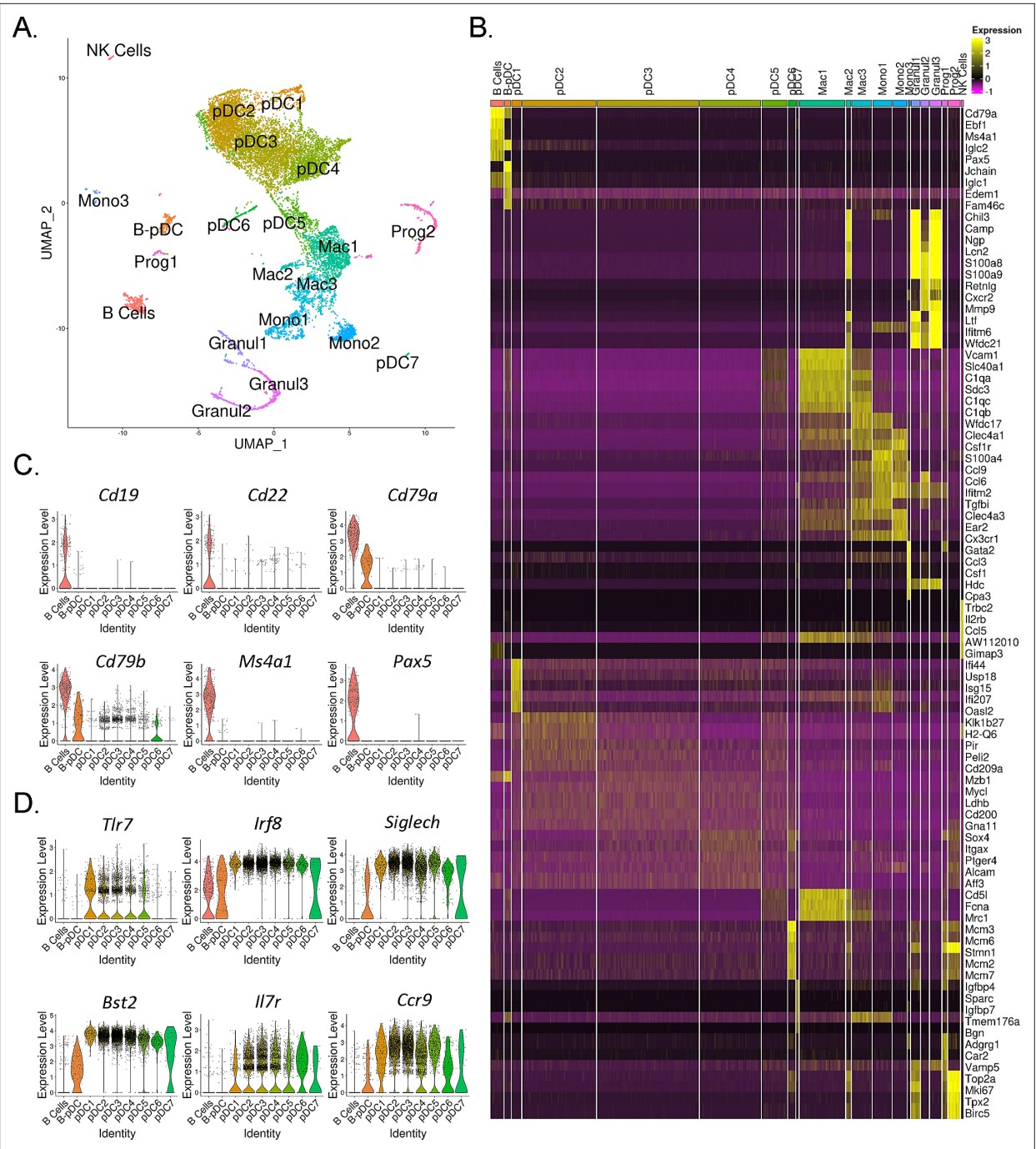

**Figure 3.** Single-cell RNA-seq analysis of mouse PDCA1⁺ bone marrow cells. (**A**) UMAP generated by Seurat clustering analysis. Cluster identities were assigned using the top 5 consensus CIPR (Cluster Identity Predictor) identity scores. (**B**) Heatmap of the top 5 differentially expressed genes (DEGs) from each cluster. After Seurat clustering, cell reads were subsetted to include only cells classified as B cells, plasmacytoid dendritic cells (pDCs), and B-pDCs. DEGs were generated for the new data subset and markers associated with (**C**) mature B cells and (**D**) classical pDCs were plotted as violin plots.

The online version of this article includes the following figure supplement(s) for figure 3:

**Figure supplement 1.** Confirmation of unbiasedly identified cell cluster identities.

cell genes, including *Jchain, Iglc1, Iglc2, Igha, Igkc* (***Supplementary file 2***). Direct DEG comparison between Prog1 and Prog2 clusters confirmed Prog1 cells exhibit a more 'Pre/Pro B cell-like' developmental profile and low expression of markers that are 'pDC-biased' relative to Prog2 (***Supplementary file 2***). After clustering and profiling PDCA1⁺ cells, we subsetted the data in order to compare B-pDCs

gene expression to only B cell and pDC clusters using Wilcoxon rank sum tests. We plotted several genes commonly associated with B cells and pDCs. Except for *Cd79a* and *Cd79b*, B-pDCs showed negligible expression of mature B cell associated genes such as *Pax5*, *Cd19*, and *Cd22* (*Figure 3C*). Expression of canonical pDC genes was present but reduced in B-pDCs relative to classical pDC subsets (*Figure 3D*). Our analysis highlights the B-pDC cluster uniqueness among pDC subsets and further supported its lineage divergence from mature B cells.

In order to temporally examine lymphoid transcriptional commitment to B cell or B-pDC lineages, we inferred developmental trajectories for B and B-pDC single cells. Pseudotime is a measure of

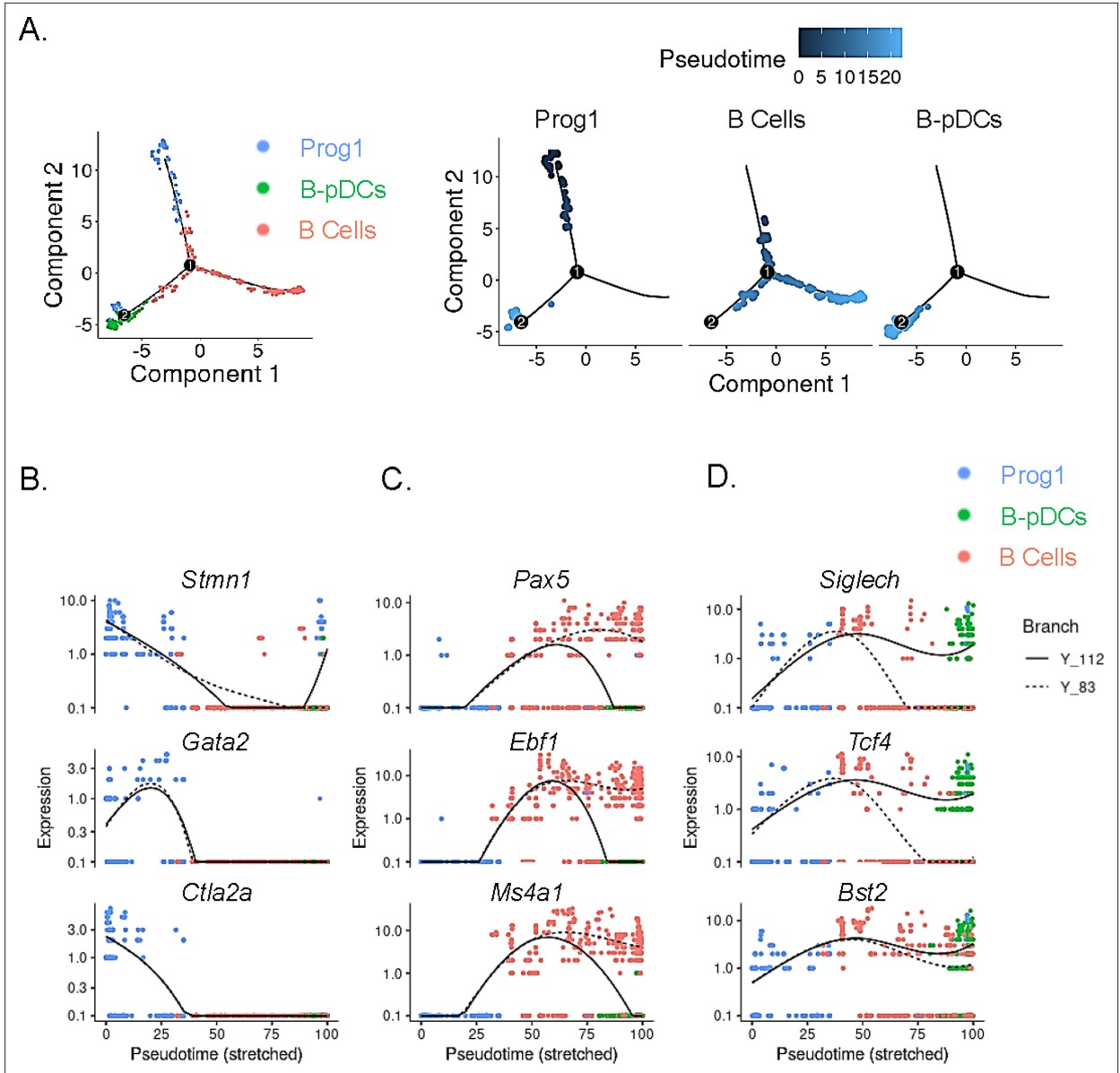

**Figure 4.** Single-cell trajectory analysis of Prog1, B cells, and B-plasmacytoid dendritic cell (B-pDC) clusters. (**A**) Single cells belonging to clusters Prog1, B cells, and B-pDCs were ordered and plotted as a function of pseudotime based on uniquely expressed markers using the unsupervised Monocle dpFeature. Cluster Prog1 was classified as the root of the trajectory given its high expression of pre-pro B cell and stem cell associated markers. Differentially expressed genes (DEGs) of interest were plotted as a function of pseudotime in Prog1 (**B**), B cells (**C**), or pDCs (**D**) using Monocle. Branch Y_112 represents the expression kinetic trend of the B-pDC cluster (green), while branch Y_83 represents the expression kinetic trend in expression of B cells according to branched expression analysis modeling, or BEAM.

The online version of this article includes the following figure supplement(s) for figure 4:

**Figure supplement 1.** Expression of early B cell receptor genes in B-plasmacytoid dendritic cells (B-pDCs).

how much progress an individual cell has made through a process such as differentiation (*Qiu et al., 2017*; *Cao et al., 2019*). In other words, cells that belong to a single cluster can be segregated across various transcriptional 'states' based on number of reads per cell of a particular set of developmental genes. Although we cannot say the pre/pro B cell-like Prog1 cluster represents an immediately adjacent common progenitor for B cells and B-pDCs, we confirmed B-pDCs are not a direct precursor for B cells based on their single-cell developmental trajectory. Cells in the Prog1 cluster were chosen as state '0' based on their abundant expression of lymphoid progenitor associated genes (*Figure 4A*). Analysis of uniquely expressed genes showed Prog1 downregulated several multipotency markers such as *Stmn1*, *Gata2*, and *Ctla2a*, while B cells increased expression of transcription factors needed for B cell commitment (e.g. *Pax5*, *Ebf1*, and *Ms4a1*) (*Figure 4B and C*). Although B-pDCs did not express detectable levels of most B cell commitment markers evaluated, they did retain expression of early BCR associated genes, including *Ly6d*, *Ighm*, *Igkc*, *Iglc2*, and *Jchain* (*Figure 4—figure supplement 1*). Because these early B cell-related markers are also expressed in plasma cells, we performed flow cytometric analysis of B220[+] PDCA1[+]:*Cd79a*[+] cells. We discovered that B-pDCs express negligible levels of TACI (a transcription factor that silences the B cell developmental program; *Shapiro-Shelef and Calame, 2005*), and that expression of CD138 and IgG1 was also negligible (*Figure 4—figure supplement 1*, and not shown), thus ruling out any major plasma cell overlap at the protein level. Lineage analysis also revealed that B-pDCs enhanced expression of the pDC associated markers *Siglech*, *Tcf4*, *Bst2*, and also of *Xbp1* (a marker expressed at high levels in secretory cells). Committed B cells in turn downregulated these markers as they progressed through pseudotime (*Figure 4D*). In summary, lineage analyses revealed that as B-pDCs develop, they retain expression of both early BCR genes and pDC associated genes, diverging from B cells through continued suppression of transcription factors required for terminal B cell differentiation.

In order to determine if murine B-pDCs are homologous with AXL[+] human DCs, we subsetted pDCs clusters and plotted their relative expression of AXL[+] DC associated genes. We identified B-pDCs and pDC5 as 'noncanonical pDCs' due to their distinguishing high expression of *Axl* (*Figure 5A*). Accordingly, *Axl* negative pDCs are referred hereafter as 'classical pDCs'. We then analyzed the top DEGs shared between B-pDCs and pDC5s using the function 'FindConservedMarkers'. *Lyz2*, the complement genes *C1qa*, *C1qb*, *C1qc*, and *Cd5l,* were the top conserved markers for these two pDC subsets (*Figure 5A*). Of note, *Cd5l*, a macrophage associated secreted protein, has been shown to play a major role in initiating inflammation and maintaining humoral responses (*Sanjurjo et al., 2015*; *Martinez et al., 2014*; *Oskam et al., 2023*). Altogether, our analysis revealed that constitutive expression of *Axl* and innate activation markers are present in more than one transcriptionally distinct murine pDC profile, stressing the functional heterogeneity and plasticity of murine pDCs.

Noncanonical AXL[+] DCs in humans have been shown to be derived from a myeloid pro-pDC progenitor (*Dress et al., 2019*; *Herman and Grün, 2018*; *Rodrigues et al., 2018*). In mice, Feng and colleagues used clonal lineage to show that pDCs and cDCs originate from a common Flt3-dependent pathway of differentiation originating from transient CX3CR1[+] progenitors. Since the authors state that their finding does 'not completely rule out the common origin of pDCs and B cells, which may bifurcate earlier in hematopoiesis', we set out to corroborate the hematopoietic origins of pDC5 and B-pDCs. We first calculated pDC5's 'community connectedness' (*Wolf et al., 2019*) to other pDC clusters and plotted their developmental trajectories originating from the pro-pDC progenitor cluster Prog2 using Monocle3 (*Trapnell et al., 2014*). Partitioning Prog2 and pDC single cells into Louvain communities yielded five supergroups, with the largest group encompassing most pDC clusters, including pDC5. The other groups consisted of non-pDCs (Prog2 cells), B-pDCs, and a few single cells (>20 cells) genotypically distinct from their original pDC clusters (*Figure 5B*). Ultimately, this statistical grouping confirmed that B-pDCs were not developmentally linked to 'classical' AXL negative pDCs or Axl[+] pDC5 cells. Accordingly, B-pDCs did not integrate into the predicted lineage trajectory originating from Prog2, whereas the *Axl*[+] pDC5 population did (*Figure 5B*). Both B-pDCs and pDC5s exhibited a similar late developmental pseudotime, while clusters pDC6 and pDC4 were the first to develop from Prog2. Collectively, our data establish that murine noncanonical pDCs (AXL[+]) consist of at least two major pDC subpopulations, which develop from lymphoid and myeloid progenitors, respectively.

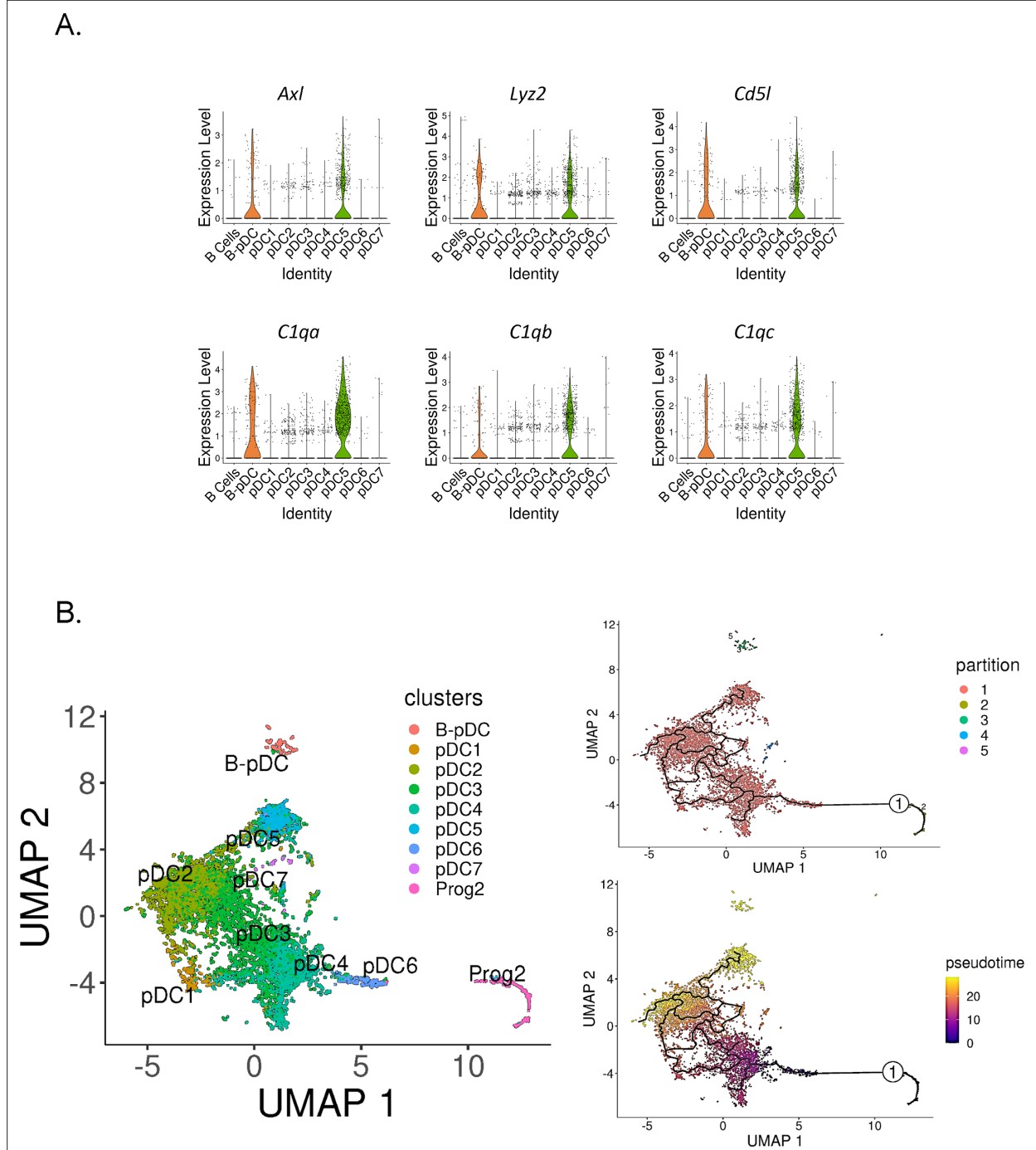

**Figure 5.** Identification of Axl⁺ noncanonical plasmacytoid dendritic cells (pDCs) in mice. (**A**) Violin plots depicting relative expression of conserved 'non canonical *AXL*⁺ DCs' associated markers in mouse pDC clusters. (**B**) Bone marrow (BM) scRNA-seq data was subsetted to include only pDCs and Prog2 subsets. We used Monocle3 for partition-based learning of cell developmental trajectories. Louvain partitions were generated from single-cell reads and their developmental trajectory was inferred using SimplePPT (top right quadrant). Pseudotime across clusters is shown in the lower bottom quadrant.

## Discussion

Dendritic cell-subset biology, development, and the ensuing nomenclature have long been unclear and everchanging. Here, we provide definitive evidence in support of the long-suspected 'lymphoid past' of pDCs by establishing their ability to arise in vivo from CLP-derived progenitors with B/pDC bipotential lineage capacity (*Pelayo et al., 2005*; *Sathe et al., 2013*; *Shigematsu et al., 2004*; *Björck*

*and Kincade, 1998*; *Izon et al., 2001*). Our data shows that the murine pDC compartment is bipartite, being comprised of B-pDCs—diverted from the CLP post T-B bifurcation—as well as myeloid-derived classical pDCs. Unlike most myeloid-derived pDCs, B-pDCs constitutively express high levels of multiple innate activation markers, including *C1qa, C1qb, Lyz2*, and *Cd81*. Functionally, they expand more readily after TLR9 engagement than classical pDCs (either through increased proliferation or differentiation of other cell types) and excel at activating T cells in culture. While further functional definition awaits discovery, our work provides a framework for the identification and segregation of the B-pDC lineage (comprising almost one-fifth of the total pDC compartment) from other myeloid-derived pDC subpopulations. Primarily, our observations support the hypothesis that DC functionality derives primarily from ontogeny rather than from tissue environment (*Heidkamp et al., 2016*), exemplified by evolution of a specialized pDC lineage from a lymphoid progenitor. In the case of the B-pDCs, our data suggest such a cell may deviate from B cell commitment after *Ly6d* expression, thought to be the earliest marker of B cell specification (*Inlay et al., 2009*).

Most noticeably, our newly described B-pDC population expresses high levels of the tyrosine kinase AXL and specializes in T cell activation through displaying high levels of MHC II and costimulatory markers. Of note, ablation of AXL in mice has previously been shown to increase expression of type I IFN while impairing IL-1β production and T cell activation during viral infections (*Schmid et al., 2016*; *Zhang et al., 2022*). Our related findings that AXL+ B-pDCs excel at T cell priming but exhibit reduced IFNα expression might suggest these DCs play an important role in advancing T cell expansion, while curbing innate immune responses that can result in autoimmune damage. Of note, whether induction of IFN-I production in vivo could also affect CLP and increase the amount of YFP+ lymphoid progenitors and thus B-pDC output is unclear. Further research is required to answer this question.

In addition to describing a novel B cell-like pDC population, we profiled murine can develop in parallel from either myeloid or lymphoid progenitors, with lymphoid-derived B-pDCs retaining expression of additional early B cell genes that are not expressed in myeloid-derived pDCs. Intriguingly, ChIP-seq analysis done by our lab of BCL11A target binding in multiple human cell lines suggests that an evolutionarily conserved transcriptional hierarchy might distinguish AXL+ pDCs and B cells in humans as well (unpublished data). The demarcation of lymphoid derived B-pDCs as one of the AXL+ pDC populations found in mice may help clarify the perceived plasticity of the pDC compartment in normal and disease contexts (*Li et al., 2017*), as well as provide a new cell for targeted study within the context of autoimmune disease, cancer, and infection models.

## Methods

### Mice

Generation of Bcl11a cKO mice (C57BL/6J background) was performed as described *Ippolito et al., 2014*. To generate conditional knockouts for the *Cd79a* gene, specifically, Bcl11a F/F mice were crossed to *Cd79a*-Cre+ Rosa26-YFP+ reporter mice. *Cd79a*-Cre deleter and Rosa26-YFP reporter strains (C57BL/6J genetic background) were obtained from Jackson labs (catalog # 020505 and 006148, respectively), bred, and genotyped according to the vendor protocol and their validated primers. All housing, husbandry, and experimental procedures with mice were approved by the Institutional Animal Care and Use Committees at the University of Texas at Austin (Protocol # AUP-2018-00051). 6-week-old gender matched male or female mice were used for all experiments described in this manuscript. For each experiment, at least four samples per experimental group were used, with each experiment repeated at least three times (except for BM reconstitutions and imaging flow cytometry experiments, which were repeated twice).

### Tissue processing

Mouse femurs were cut at the extremities and a 25G needle was inserted in the bones to flush out BM cells with 10 mL of cold PBS onto a 70 μM strainer in a 50 mL tube. Cells were washed from the strainers into the tubes with an additional 10 mL of PBS. Tubes were spun at 300G for 5 min and supernatant was decanted. 2 mL of ACK red blood cell lysis buffer was used to resuspend cells, which were incubated for 2 min and 20 s. Samples were diluted with 20 mL of cold PBS, washed once, re-filtered and resuspended in 200 μL of FACS buffer for subsequential flow cytometry staining. Spleens were mashed onto pre-wet 70 μM filters in a 50 mL tube with syringe plungers and 20 mL of cold PBS was

added through the strainers into the tube. Cells were spun at 300G for 5 min and supernatant was decanted. 2 mL of ACK red blood cell lysis buffer was used to resuspend cells, which were incubated for 2 min and 20 s. Samples were diluted with 20 mL of cold PBS, washed once, re-filtered, and resuspended in 200 μL of FACS buffer for subsequential flow cytometry staining.

## Flow cytometry

Single-cell suspensions ($1 \times 10^6$ cells) were resuspended in 200 μL of FACS buffer (5 mM EDTA and 1% Heat Inactivated Fetal Calf Serum in D-PBS) and incubated for 15 min with surface antibody cocktail (each antibody at a concentration of 1.25 μg/mL in FACS buffer) in the dark at 4°C. Cells were then washed three times with FACS buffer and ran live on a BD LSRII Fortessa. Single color controls for panel compensation were made using AbC Total Antibody Compensation Bead kit (Thermo Fisher Scientific), by staining each control with one of the panel antibodies and using these controls to compensate the flow cytometer experiment prior to running the samples. A non-stained sample of cells was included in experimental runs and used to adjust for autofluorescence and any nonspecific staining. FMO single-cell controls were also included in some of the experimental runs (cells stained with all antibodies in the panel minus one), for compensation quality control and to detect positive MFI thresholds for each fluorophore using cells. Sorting was performed on a FACS Aria (BD Biosciences). All analysis was done using FlowJo (Tree Star) software. Single-cell suspensions were stained with the following antigen-specific monoclonal antibodies (clones in parenthesis): anti CD45R/B220-BV605 (RA3-6B2), CD19Alexa Fluor 700 (6D5), CD86-APC-Cy7 (GL-1), I-A/I-E-Pacific Blue (M5/114.15.2), IL7R-BV421 (A7R34), cKit-PE-Cy7 (ACK2), CD11b-PerCP-Cy5.5 (M1/70), CD3e-PerCP-Cy5.5 (500A2), Gr1-PerCP-Cy5.5 (RB6-8C5), CD4-PerCP-Cy5.5 (RM4-5), CD8-PerCpCy5.5 (53–6.7), NK1.1-PerCp-Cy5.5 (PK136), Sca1-BV711 (D7), CD135-APC (A2F10), CD150-BV605 (TC15-12F12.2), *Cd79a*-FITC (24C2.5) (Invitrogen), Anti-CD11c-PerCP-Cy5.5 (N418) (eBioscience), Anti-AXL-APC (175128) (R&D Systems), Anti-CD23 BV510 (B3B4) (BioLegend), Anti-CD267-BV421 (8F10) (BD Biosciences), Anti-CD138-APC (281-2) (BioLegend), Anti-CD83-PeCy7 (Michel-19), and Anti-PDCA1-PE or Biotin (JF05-1C2.4.1) (Miltenyi Biotech), in D-PBS/2% (vol/vol) FBS FACS buffer. In some experiments, viability stain Ghost Dye Red 780 (Fisher Scientific) was used to stain cells in PBS prior to antibody staining according to the manufacturer's protocol. Flow cytometry data was analyzed with FlowJo, while imaging flow cytometry was performed on ImageStream X and analyzed on IDEAS 6.0 (Amnis), by pre-filtering out of focus cells and performing automatic software guided compensation with single color controls.

## BM transplantation

BM from 6-week-old Bcl11aF/F/*Cd79a*-Cre+cKO or Mb-1-Cre-YFP reporter mice (n=3 per group) were collected from femurs and $5 \times 10^6$ BM cells were transferred via retro-orbital injection into recipient immunocompetent C57BL/6J mice lethally irradiated with two doses of 450 rad 1 hr apart. Mice were kept on an antibiotic diet for 3 weeks to allow for immune reconstitution. Eight weeks post transfer, mice were sacrificed and cells collected from BM and spleen to investigate reconstitution of cellular subsets.

## RNA isolation and RNA-seq

For bulk RNA-seq, 12 mice were pooled into three YFP$^+$ groups, and total RNA was extracted as above. Oligo-dT-primed cDNA was prepared using SuperScript III First-Strand Synthesis System for RT-PCR (Invitrogen). Poly(A) mRNA was enriched using the NEBNext magnetic isolation module (E7490) and samples underwent DNAse treatment. cDNA was prepared using the Ultra Low kit from Clontech (Mountain View, CA, USA). Libraries were prepared according to the manufacturer's instructions for the NEBNext Ultra II Direction RNA kit (NEB, product number E7760). The resulting libraries tagged with unique dual indices were checked for size and quality using the Agilent High Sensitivity DNA Kit (Agilent). Library concentrations were measured using the KAPA SYBR Fast qPCR kit and loaded for sequencing on the Illumina NovaSeq 6000 instrument (paired-end 2×150, or single-end, 100 cycles; $30 \times 10^6$ reads/sample). Data were analyzed using a high-throughput next-generation sequencing analysis pipeline: FASTQ files were aligned to the mouse genome (mm9, NCBI Build 37) using TopHat2[53]. Gene expression profiles for the individual samples were calculated with Cufflinks *Trapnell et al., 2013* as RPKM values. YFP$^+$ pDC samples were normalized to each control YFP$^-$ pDC sample. GO terms identified as significantly different: GO:0006955, GO:0009611, GO:0048584, GO:0006954,

GO:0002684, GO:0001775, GO:0009986. Panther terms: BP00148, BP00155, BP00102, BP00120. GSEA using ordered gene expression levels of CLP-derived B-pDC and CMP-derived pDC were significantly enriched in both GO terms and Panther-derived gene sets. A randomly selected control GSEA curated set, GSE7831: UNSTIM_VS_ INFLUENZA_STIM_ PDC_4H_DN (defined as genes down-regulated in untreated pDC vs. influenza virus infected pDC 55) showed insignificant enrichment. Normalized enrichment score and false discovery rate q-values (FDR): FDR≤0.25 is considered significant *Mootha et al., 2003*.

## TLR9 engagement

*Cd79a*-cre-YFP reporter mice were injected via tail veins with 100 µL of PBS containing 50 µg of CpG:ODN (n=4) or GpC:ODN control (n=4) (CpG-B no. 1826, TCCAT GACGTTCCT GACGTT; control non-CpG-B no. 2138, TCCATGAGCTTCCTGAGCTT, Invivogen, USA). At 24 hr, mice were sacrificed and BM and spleens were collected for pDC phenotyping.

## Cell co-cultures

*Cd79a*-YFP$^+$ B-pDC (B220$^+$ PDCA1$^+$YFP$^+$) and *Cd79a*-YFP$^-$ pDCs (B220$^+$PDCA1$^+$YFP$^-$) subsets were sorted on a FACS Aria (BD Biosciences) and cultured in 96-well round-bottom plates (n=3 wells in triplicates, repeated twice) with RPMI medium 1640 containing 10% (vol/vol) FBS, 2 mM L-glutamine, 100 units/mL penicillin and streptomycin, 1 mM sodium pyruvate, and 10 mM HEPES, and with or without 5 µg/mL class A CpGs (ODN 1585, Invivogen) for 24 hr. pDCs were then washed three times to remove residual CpG. Preactivated pDCs (5×10$^3$) were cultured with lymphocytes (2.5×10$^4$) magnetically sorted from mouse spleen single-cell preparations using MACs columns and the EasySep Mouse T Cell Isolation Kit (StemCell Technologies Cat# 19851) for a total of 6 d with 20 units/mL of IL-2. CSFE labeling of lymphocytes (for which sorting purity was determined to be 97% via flow cytometry) prior to co-culturing experiment was done using the CellTrace CFSE Cell Proliferation Kit (Thermo Fisher Scientific; Cat# C34554) according to the manufacturer's protocol.

## Enzyme-linked immunosorbent assays

Sorted B-pDC and pDCs (5×10$^3$ sorted cells per well, n=3 per group, done in duplicate wells for each condition) were stimulated with 5 µM CpG-A (ODN 1585, Invivogen) or CpG-C (ODN 2395, Invivogen); or left untreated for 24 hr. ELISAs for IFNα (Invitrogen) or IL-12p40 (BioLegend), respectively, was performed in 100 µL of cell-free supernatants according to the manufacturer's instructions. Data was acquired using a Bio-Tek model EL311 automated microplate reader measuring absorbance at 450 nm.

## Statistical analysis

All flow cytometry or ELISA data were analyzed with Prism (v8; GraphPad, La Jolla, CA, USA) using Mann-Whitney non-parametric t-tests. All bar graphs show means and standard deviation (SD) and are representative of repeated experiments. These data were considered statistically significant when p-values were <0.05. 10x single-cell RNA-seq analysis PDCA1$^+$ C57BL/6J BM cells from 4 pooled 6-week-old wildtype C57BL/6J mice were magnetically sorted using MACS columns into culture medium, washed once with PBS + 0.04% BSA, and resuspended in 32 µL PBS + 0.04% BSA. Single-cell suspensions(50,000) were processed in the University of Texas Genomic Sequencing and Analysis Facility (GSAF). Cell suspensions were loaded on the Chromium Controller (10X Genomics) and processed for cDNA library generation following the manufacturer's instructions for the Chromium NextGEM Single Cell 3' Reagent Kit v3.1 (10X Genomics). The resulting libraries were examined for size and quality using the Bioanalyzer High Sensitivity DNA Kit (Agilent) and their concentrations were measured using the KAPASYBR Fast qPCR kit (Roche). Samples were sequenced on the NovaSeq 6000 instrument (paired-end, read 1: 28 cycles, read 2: 90 cycles) with a targeted depth of 71,130 reads/cell. Cellranger (v3.0.2) was used to demultiplex samples, to map data to the mouse transcriptome (mm10) and quantify genes. The gene counts matrix was read into Seurat (v3.2.1). Cells with unique feature counts over 3500 or less than 200, and >3% mitochondrial read counts were removed from the analysis. The data was normalized and transformed using scTransform. Cells were clustered based on the top 1000 variable features, 20 PCs, and a resolution of 0.6 (Seurat's graph-based clustering approach), and then visualized using UMAP. Analysis of the normalized and filtered single-cell

gene expression data (median of 16,181 genes across 13,838 single-cell transcriptomes) was used for the final expression file and downstream analysis. Wilcoxon rank sum test was used to test for differential expression among clusters and identify gene markers (logfc.threshold=1, min.pct=0.25). Gene markers and other genes of interest were visualized using violin plots, dot plots, and heatmap plots. Unsupervised ordering of PDCA1$^+$ cells was done with the Seurat integrated results as input to build a tree-like differentiation trajectory using the DDRTree algorithm of the R package Monocle v2 59. Select DEGs of B-pDCs, B cells, and Prog1 populations were set as ordering genes for each trajectory, with the root state set as Prog1 classified cells. Expression of genes as a function of pseudotime was plotted with the plot_genes_branched_pseudotime Monocle function. We also used Monocle3 to generate partition-based learning of single-cell developmental trajectories with SimplePPT and plot UMAP-based pseudotime figures with initial reclustering parameters num_dim = 60, resolution = 1e-5.

## Acknowledgements

We thank June V Harriss for expert assistance in the generation of Bcl11a conditional knockout mice, and Chhaya Das and Maya Ghosh for help in ChIP experiments and cell culture. The CAL1 cell line was kindly provided by Drs. Takahiro Maeda and Boris Reizis. Library preparation and Illumina ChIP- and RNA-seq were performed at the NGS core of the MD Anderson Cancer Center. Single-cell RNA-seq was performed at the University of Texas Genomic Sequencing and Analysis Facility with the help of Holly Stevenson and Dhivya Arasappan. The Lymphoma Research Foundation Fellowship 300463 (to JDD); NIH grant F32CA110624 and Owens Medical Research Foundation grant (to GCI); NIH grant R35GM133658 (to SY); NIH grant R01AI104870 (to LIRE); NIH grant R01CA130075 (to VRI); NIH Grant R01CA31534; Cancer Prevention Research Institute of Texas (CPRIT) Grant RP120348 to the MD Anderson NGS core and CPRIT RP120459 and the Marie Betzner Morrow Centennial Endowment (to HOT) provided support for this work.

## Additional information

### Funding

| Funder | Grant reference number | Author |
| --- | --- | --- |
| NIH Office of the Director | Ruth L. Kirschstein Postdoctoral Individual National Research Service Award F32CA110624 | Gregory Ippolito |
| NIH Office of the Director | Maximizing Investigators' Research Award (MIRA) (R35) R35GM133658 | Stephen Yi |
| NIH Office of the Director | NIH Research Project Grant Program (R01) R01AI104870 | Lauren IR Ehrlich Vishwanath R Iyer |
| Lymphoma Research Foundation | The Lymphoma Research Foundation Fellowship 300463 | Joseph D Dekker |
| NIH Office of the Director | NIH Research Project Grant Program (R01) R01CA130075 | Vishwanath R Iyer |
| Cancer Prevention and Research Institute of Texas | CPRIT RP120459 | Gregory Ippolito |
| Marie Betzner Morrow Centennial Endowment | Endowment | Haley O Tucker |
| National Institutes of Health | | Haley O Tucker |
| CPRIT | RP120459 | Haley O Tucker |

| Funder | Grant reference number | Author |
|--------|------------------------|--------|

The funders had no role in study design, data collection and interpretation, or the decision to submit the work for publication.

## Author contributions

Alessandra Machado Araujo, Software, Formal analysis, Investigation, Methodology, Writing – original draft, Writing – review and editing; Joseph D Dekker, Resources, Data curation, Software, Formal analysis, Validation, Investigation, Visualization, Methodology, Writing – original draft, Project administration, Writing – review and editing; Kendra Garrison, Catherine Rhee, Zicheng Hu, Bum-Kyu Lee, Jiwon Lee, Vishwanath R Iyer, Investigation; Zhe Su, Software, Investigation, Methodology; Daniel Osorio, Resources, Methodology; Lauren IR Ehrlich, George Georgiou, Writing – original draft, Writing – review and editing; Gregory Ippolito, Haley O Tucker, Conceptualization, Resources, Data curation, Software, Formal analysis, Supervision, Funding acquisition, Validation, Investigation, Visualization, Methodology, Writing – original draft, Project administration, Writing – review and editing; Stephen Yi, Supervision, Methodology

## Author ORCIDs

Alessandra Machado Araujo (iD) https://orcid.org/0000-0001-7568-074X
Joseph D Dekker (iD) https://orcid.org/0000-0002-2068-3529
Vishwanath R Iyer (iD) https://orcid.org/0000-0003-3383-248X
Lauren IR Ehrlich (iD) https://orcid.org/0000-0002-1697-1755
Haley O Tucker (iD) https://orcid.org/0000-0001-7735-2862

## Ethics

All housing, husbandry, and experimental procedures with mice were approved by the Institutional Animal Care and Use Committees at the University of Texas at Austin. (Protocol # AUP-2018-00051).

Reviewer #3 (Public Review): https://doi.org/10.7554/eLife.96394.3.sa1
Author response https://doi.org/10.7554/eLife.96394.3.sa2

---

# Additional files

## Supplementary files

• Supplementary file 1. Top CIPR (Cluster Identity Predictor) assigned identity scores for Seurat cluster differentially expressed genes (DEGs). CIPR IDs were generated using the top 1000 DEGs from each cluster. The top 5 consensus CIPR ID generated was used to rename Seurat clusters.

• Supplementary file 2. Differentially expressed gene (DEG) comparison between progenitor clusters. The top 50 differentially expressed features of Prog1 in relation to Prog2 were calculated using Seurat using non-parametric Wilcoxon rank sum test. In blue are upregulated genes and in red are downregulated genes.

• MDAR checklist

## Data availability

RNA-seq: GSE105827; ChIP-seq: GSE99019; 10x ScRNA-seq: GSE225768. Accession Numbers to previously published data sets: GSE52868 (pre-B RNA-seq).

The following datasets were generated:

| Author(s) | Year | Dataset title | Dataset URL | Database and Identifier |
|-----------|------|---------------|-------------|------------------------|
| Araujo AM, Dekker JD, Garrison K, Su Z, Rhee C, Hu Z, Lee BK, Hurtado DS, Lee J, Iyer VR, Elrlich LIR, Georgiou G, Ippolito GC, Stephen YS, Tucker HO | 2018 | Common lymphoid progenitor derivation of plasmacytoid dendritic cells is mediated by Bcl11a | https://www.ncbi.nlm.nih.gov/geo/query/acc.cgi?acc=GSE105827 | NCBI Gene Expression Omnibus, GSE105827 |
| Araujo AM, Dekker JD, Garrison K, Su Z, Rhee C, Hu Z, Lee BK, Hurtado DO, Lee J, Iyer VR, Ehrlich LIR, Georgiou G, Ippolito GC, Stephen YS, Tucker HO | 2017 | Common lymphoid progenitor derivation of plasmacytoid dendritic cells is mediated by Bcl11a | https://www.ncbi.nlm.nih.gov/geo/query/acc.cgi?acc=GSE99019 | NCBI Gene Expression Omnibus, GSE99019 |
| Araujo AM, Dekker JD, Garrison K, Su Z, Rhee C, Hu Z, Lee BK, Hurtado DO, Lee J, Iyer VR, Ehrlich LIR, Georgiou G, Ippolito GC, Stephen YS, Tucker HO | 2023 | Common lymphoid progenitor derivation of plasmacytoid dendritic cells is mediated by Bcl11a | https://www.ncbi.nlm.nih.gov/geo/query/acc.cgi?acc=GSE225768 | NCBI Gene Expression Omnibus, GSE225768 |

The following previously published dataset was used:

| Author(s) | Year | Dataset title | Dataset URL | Database and Identifier |
|-----------|------|---------------|-------------|------------------------|
| Liu G, Hu Y, Smyth GK, Dickins R | 2014 | Expression profiling of mouse bone marrow pre-B cells | https://www.ncbi.nlm.nih.gov/geo/query/acc.cgi?acc=GSE52868 | NCBI Gene Expression Omnibus, GSE52868 |

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
