## [Editor Report · eLife assessment]

This **valuable** study reports that while most plasmacytoid dendritic cells (pDCs) originate from common dendritic cell precursors, approximately 20% are derived from lymphoid progenitors shared with B cells. The methodology used and the evidence are **solid**, and further demonstrate the distinct transcription factor requirements and activities of this subset of pDCs, although the functional significance of this dendritic cell subset will require further elucidation. The findings will be of great interest for those interested in the developmental and functional biology of the immune system.

---

## [Referee Report · Reviewer #3 (Public Review)]

Summary:

Plasmacytoid dendritic cells (pDCs) represent a specialized subset of dendritic cells (DCs) known for their role in producing type I interferons (IFN-I) in response to viral infections. It was believed that pDCs originated from common DC progenitors (CDP). However, recent studies by Rodrigues et al. (Nature Immunology, 2018) and Dress et al. (Nature Immunology, 2019) have challenged this perspective, proposing that pDCs predominantly develop from lymphoid progenitors expressing IL-7R and Ly6D. A minor subset of pDCs arising from CDP has also been identified as functionally distinct, exhibiting reduced IFN-I production but a strong capability to activate T cell responses. On the other hand, clonal lineage tracing experiments, as recently reported by Feng et al. (Immunity, 2022), have demonstrated a shared origin between pDCs and conventional DCs (cDCs), suggesting a contribution of common DC precursors to the pDC lineage.

In this context, Araujo et al. investigated the heterogeneity of pDCs in terms of both development and function. Their findings revealed that approximately 20% of pDCs originate from lymphoid progenitors common to B cells. Using Mb1-Cre x Bcl11a floxed mice, the authors demonstrated that the development of this subset of pDCs, referred to as "B-pDCs," relied on the transcription factor BCL11a. Functionally, B-pDCs exhibited a diminished capacity to produce IFN-I in response to TLR9 agonists but secreted more IL-12 compared to conventional pDCs. Moreover, B-pDCs, either spontaneously or upon activation, exhibited increased expression of activation markers (CD80/CD86/MHC-II) and a heightened ability to activate T cell responses in vitro compared to conventional pDCs. Finally, Araujo et al. characterized these B-pDCs at the transcriptomic level using bulk and single-cell RNA sequencing, revealing them as a unique subset of pDCs expressing certain B cell markers such as Mb1, as well as specific markers (Axl) associated with cells recently described as transitional DCs.

Thus, in contrast to previous findings, this study posits that a small proportion of pDCs derive from B cell-committed lymphoid progenitors, and this subset of B-pDCs exhibits distinct functional characteristics, being less specialized in IFN-I production but rather in T cell activation.

Strengths:

Previously, the same research group delineated the significance of BCL11a as a critical transcription factor in pDC development (Ippolito et al., PNAS, 2014). This study elucidates the precise stage during hematopoiesis at which BCL11a expression becomes essential for the emergence of a distinct subset of pDCs, substantiated by robust genetic evidence in vivo. Furthermore, it underscores the shared developmental origin between pDCs and B cells, reinforcing prior research in the field that suggests a lymphoid origin of pDCs. Finally, this works attributes specific functional properties to pDCs originating from these lymphoid progenitors shared with B cells, emphasizing the early imprinting of functional heterogeneity during their development.

Weaknesses:

Using their Mb1-reporter mice, the authors demonstrate that YFP pDCs originating from lymphoid progenitors are functionally distinct from conventional pDCs, mostly in vitro, but their in vivo relevance remains unknown. As underlined by both reviewers I believe that it is crucial to investigate how Bcl11a conditional deficiency in Mb1 expressing cells affects the anti-viral immune response, for example, using the M-CoV infection model as described by Sulczewski et al. in Nature Immunology, 2023. The current in vivo data using TLR9 agonist and in vitro data using B-pDCs co-cultures with T cells insufficiently address what B-pDCs might be doing in infectious contexts.

Revisions:

I thank the authors for their responses to my questions and for addressing most of my comments clearly and thoroughly. However, one major question remains unanswered: What is the functional relevance of the subset of B-pDCs that they have characterized? This key question, also highlighted by the other reviewer, requires further investigation. The current in vivo data using TLR9 agonist and in vitro data using B-pDCs co-cultures with T cells insufficiently address what B-pDCs might be doing in infectious contexts.

---

## [Author Response]

The following is the authors’ response to the original reviews.

**Public Reviews:**

**Reviewer #1 (Public Review):**
Summary:Plasmacytoid dendritic cells (pDCs) represent a specialized subset of dendritic cells (DCs) known for their role in producing type I interferons (IFN-I) in response to viral infections. It was believed that pDCs originated from common DC progenitors (CDP). However, recent studies by Rodrigues et al. (Nature Immunology, 2018) and Dress et al. (Nature Immunology, 2019) have challenged this perspective, proposing that pDCs predominantly develop from lymphoid progenitors expressing IL-7R and Ly6D. A minor subset of pDCs arising from CDP has also been identified as functionally distinct, exhibiting reduced IFN-I production but a strong capability to activate T-cell responses. On the other hand, clonal lineage tracing experiments, as recently reported by Feng et al. (Immunity, 2022), have demonstrated a shared origin between pDCs and conventional DCs (cDCs), suggesting a contribution of common DC precursors to the pDC lineage.In this context, Araujo et al. investigated the heterogeneity of pDCs in terms of both development and function. Their findings revealed that approximately 20% of pDCs originate from lymphoid progenitors common to B cells. Using Mb1-Cre x Bcl11a floxed mice, the authors demonstrated that the development of this subset of pDCs, referred to as "B-pDCs," relied on the transcription factor BCL11a. Functionally, B-pDCs exhibited a diminished capacity to produce IFN-I in response to TLR9 agonists but secreted more IL-12 compared to conventional pDCs. Moreover, B-pDCs, either spontaneously or upon activation, exhibited increased expression of activation markers (CD80/CD86/MHC-II) and a heightened ability to activate T-cell responses in vitro compared to conventional pDCs. Finally, Araujo et al. characterized these B-pDCs at the transcriptomic level using bulk and single-cell RNA sequencing, revealing them as a unique subset of pDCs expressing certain B cell markers such as Mb1, as well as specific markers (Axl) associated with cells recently described as transitional DCs.Thus, in contrast to previous findings, this study posits that a small proportion of pDCs derive from B cell-committed lymphoid progenitors, and this subset of B-pDCs exhibits distinct functional characteristics, being less specialized in IFN-I production but rather in T cell activation.Strengths:Previously, the same research group delineated the significance of BCL11a as a critical transcription factor in pDC development (Ippolito et al., PNAS, 2014). This study elucidates the precise stage during hematopoiesis at which BCL11a expression becomes essential for the emergence of a distinct subset of pDCs, substantiated by robust genetic evidence in vivo. Furthermore, it underscores the shared developmental origin between pDCs and B cells, reinforcing prior research in the field that suggests a lymphoid origin of pDCs. Finally, this work attributes specific functional properties to pDCs originating from these lymphoid progenitors shared with B cells, emphasizing the early imprinting of functional heterogeneity during their development.Weaknesses:The authors delineate a subset of pDCs dependent on the BCL11a transcription factor, originating from lymphoid progenitors, and compare it to conventional pDCs, which they suggest differentiate from common DC progenitors of myeloid origin. However, this interpretation lacks support from the authors' data. Their single-cell RNA sequencing data identifies cells corresponding to progenitors (Prog2), from which the majority of pDCs, termed conventional pDCs, likely originate. This progenitor cell population expresses Il7r, Siglech, and Ly6D, but not Csfr1. The authors describe this progenitor as resembling a "pro-pDC myeloid precursor," yet these cells align more closely with lymphoid (Il7r+) progenitors described by Rodrigues et al. (Nature Immunology, 2018) and Dress et al. (Nature Immunology, 2019). Furthermore, analysis of their Mb1 reporter mice reveals that only a fraction of common lymphoid progenitors (CLP) express YFP, giving rise to a fraction of YFP+ pDCs. However, this does not exclude the possibility that YFP- CLP could also give rise to pDCs. The authors could address this caveat by attempting to differentiate pDCs from both YFP+ and YFP- CLPs in vitro in the presence of FLT3L. Additionally, transfer experiments using these lymphoid progenitors could be conducted in vivo to assess their differentiation potential in competitive settings.

Dear Reviewer 1, we appreciate your thoughtful comments. We made the decision to address the Prog2 cluster as “pro-pDC myeloid precursor” because despite its lack of CSFR-1, its CIPR similarity score showed highest transcriptional similarity to the population “SC.CDP.BM” (GEO accession number: GSM791114), which is shown to be Sca1- Flt3+ cKitlo.

A similar population identified as “common dendritic cell progenitor” is shown by Onai and colleagues (Onai et al. 2013, Immunity) to be capable of differentiating into pDCs by upregulating E2-2 and subsequently downregulating M-CSFR. In addition, we were unable to infer a developmental trajectory between Prog2 and B-pDCs using SimplePPT on Monocle3 (Figure 5B). Since we know our B-pDCs are CLP derived and most likely share a B cell progenitor population, we feel this lack of connectivity to the UMAP myeloid partition corroborates our assignment of Prog2 as a myeloid pDC progenitor (not CLP derived). Of note, recent work by Medina and colleagues has shown that while IL-7Rα knockout mice exhibit a block in B cell development at the all-lymphoid progenitor (ALP) stage, PDCA-1+ pDCs identified within the initially gated BLP population persisted (PLoS One, 2013), suggesting the IL7R chain is not required for the development of PDCA1+ cells.

Using their Mb1-reporter mice, the authors demonstrate that YFP pDCs originating from lymphoid progenitors are functionally distinct from conventional pDCs, mostly in vitro, but their in vivo relevance remains unknown. It is crucial to investigate how Bcl11a conditional deficiency in Mb1-expressing cells affects the anti-viral immune response, for example, using the M-CoV infection model as described by Sulczewski et al. in Nature Immunology, 2023. Particularly, the authors suggest that their B-pDCs act as antigen-presenting cells involved in T-cell activation compared to conventional pDCs. However, these findings contrast with those of Rodrigues et al., who have shown that pDCs of myeloid origin are more effective than pDCs of lymphoid origin in activating T-cell responses. The authors should discuss these discrepancies in greater detail. It is also notable that B-PDCs acquire the expression of ID2 (Figure S3A), commonly a marker of conventional/myeloid DCs. The authors could analyze in more detail the acquisition of specific myeloid features (CD11c, CX3CR1) by this B-PDCs subset and discuss how the expression of ID2 may impair classical pDC features, as ID2 is a repressor of E2-2, a master regulator of pDC fate.

Both reviewers expressed the need to further investigate how Bcl11a conditional deficiency in Mb1-expressing cells affects anti-viral responses of B-pDCs. While the functional characterization of B-pDC in the context of infection could be highly informative, it is really outside the scope of the present study. Our discovery that B-pDCs expand robustly upon TLR-9 agonist challenges in vivo and can prime T cells in vitro efficiently, however, suggests that these cells might play an important role during viral infections or anti-cancer immunity.

Finally, through the analysis of their single-cell RNA sequencing data, the authors show that the subset of B-pDCs they identified expresses Axl, confirmed at the protein level. Given this specific expression profile, the authors suggest that B-pDCs are related to a previously described subset of transitional DCs, which were reported to share a common developmental path with pDCs, (Sulczewski et al. in Nature Immunology, 2023). While intriguing, this observation requires further phenotypic and functional characterization to substantiate this claim.

We agree with the reviewer’s comments. We are currently preparing a separate manuscript addressing the commonalities between human transitional DCs and murine non-conventional pDCs.

**Reviewer #2 (Public Review):**
Summary:The origin of plasmatoid dendritic cells and their subclasses continues to be a debated field, akin to any immune cell field that is determined through the expression of surface markers (relative to clear subclass separation based on functional biology and experimentation). In this context, in this manuscript by Araujo et al, the authors attempt to demonstrate that a subtype of pDCs comes from lymphoid origin due to the presence of some B cell gene expression markers. They nomenclature these cells as B-pDCs. Strikingly, pDCs function via expression of IFNa where as B-pDCs do not express IFNa - thereby raising the question of what are their physiological or pathophysiological properties. B-pDCs also express AXL, a marker not seen in mouse pDCs but observed in human pDCs. Overall, using a combination of gene expression profiling of immune cells isolated from mice via RNA-seq and single-cell profiling the authors propose that B-pDCs are a novel subtype of pDCs in mice that were not previously identified and characterized.Weaknesses:My two points of discussion about this manuscript are as follows.(1) How new are these observations that pDCs could also originate from common lymphoid progenitors. This fact has been previously outlined by many laboratories including Shigematsu et al, Immunity 2004. These studies in the manuscript can be considered new based on the single-cell profiling presented, only if the further characterization of the isolated B-pDCs is performed at the functional biology level. Overlapping gene expression profiles are often seen in developing immune cell types- especially when only evaluated at the RNA expression level- and can lead to cell type complexity (and identification of new cell types) that are not biologically and functionally relevant.

Dear reviewer 2, we appreciate your thoughtful comments. We believe our single cell seq analysis adds new information to the studies mentioned because of our broader approach to BM profiling. By using only one marker (PDCA1+), scRNA-seq allowed us to dissect not only several subpopulations of pDCs that to our knowledge were not previously dissected in mice, but also linked the transcriptional similarity of B-pDCs to myeloid derived pDCs (and even other myeloid cell types), as well as B cells.

(2) The authors hardly perform any experiments to interrogate the function of these B-pDCs. The discussion on this topic can be enhanced. Ideally, some biological experiments would confirm that B-pDCs are important.Dear reviewer 2, we appreciate your thoughtful comment and agree about the need for further functional characterization of B-pDCs (please see comments directed to reviewer 1 above).(1) Considering that Bcl11a conditional deficiency severely impacts the B cell lineage, there is a possibility that such an effect on B cells may indirectly influence pDC development. To address this, the authors could repeat their bone marrow transfer experiments in a competitive setting by mixing both Bcl11a WT and CKO BM cells (using congenic markers to track the origin of the BM cells) and then specifically assess whether BM cells originating from Bcl11a CKO donors have impaired pDC output.

Dear reviewer 2, while the comment above is valid (that the reduced number of mature B cells in our Bcl11a conditional knockout might indirectly impact B-pDC development), we and many others have previously shown that lack of transcriptional regulation of E2-2 and other pDC differentiation modulators by Bcl11a (including ID2 and MTG16) intrinsically and selectively disrupts the pDC lineage. At the current stage, we feel rederiving Bcl11a cKOs and performing bone marrow transfers (which usually take several months) only to investigate indirect effects of B cells on pDC developments is outside the scope of this publication.

(2) As mentioned earlier, it is important to assess the potential of CLP, whether YFP- or YFP+, in their ability to give rise to pDCs both in vitro and in vivo. This is also crucial since the authors previously demonstrated that Bcl11a deficiency in all hematopoietic cells had a more drastic impact on pDC development than mb1-cre specific deficiency.

We agree the manuscript could be strengthened by differentiation experiments. However, in our previous publication (mentioned above by the reviewer), we specifically show that although fewer overall LSK progenitors were detected in Vav-Cre+ F/F mice, both MDP and CDP progenitor populations persisted within the Flt3+ compartment in cKO mice at percentages similar to controls. MDP (Lin– Flt3+ Sca-1− CD115+ c-kithi); CDP (Lin– Flt3+ Sca-1− CD115+ c-kitlo). This data confirms that CLPs give rise to a substantial pool of pDC subpopulations. Other works have shown this as well, both in vivo and in vitro (Wang et al. Immunity 2004; Karsunky et al, JEM 2003, etc). We therefore feel that confirming the previous observations that CLPs can give rise to pDCs is unnecessary, as our main goal in this manuscript was to describe a new pDC subpopulation that emerges primarily from CD79a+ B cell biased progenitors.

(3) The authors show a more severe impact of Bcl11a CKO on pDC depletion in the spleen than in the BM. Is this effect specific to the spleen, or can it also be observed in lymph nodes? What is the overall impact of Bcl11a conditional deficiency on pDC distribution in tissues such as the liver and lung? These questions are important to address to understand whether the heterogeneity of pDCs is differentially affected by their localization.

We agree heterogeneity of pDCs can be affected by their microenvironment. Although phenotyping of lymph nodes in Bcl11a cKOs would greatly add to our manuscript, the genetically altered strains required are no longer being bred in our facility and resurrecting them from frozen sperm is outside the realm of this publication.

(4) Regarding the functional study of pDCs, as emphasized previously, it is important to assess the in vivo relevance of B-pDCs in infectious settings.

Dear reviewer 2, we appreciate your thoughtful comment. Please see our response directed to reviewer 1 above.

(5) The authors injected CpG-ODN into mice and analyzed pDC phenotype upon activation. It is important to note that upon activation, especially upon induction of IFN-I production in vivo, mPDCA1 expression is no longer specific to pDCs (Blasius et al, Journal of Immunology, 2006). Therefore, to specifically characterize pDC phenotype upon activation, a differential gating strategy is required (CD11c, B220, Ly6C, and Siglec H) to ensure that bona fide pDCs are analyzed.

We agree with the reviewer that this would be a more appropriate characterization. Regarding PDCA1 promiscuity in activated states, we are not aware of any cell types that express very high levels of B220 and PDCA1 simultaneously other than pDCs. We therefore firmly believe that our assignment is valid. Interestingly, gating B220+ cells of Cpg challenged mice that show intermediate expression of PDCA1 results in an increase in the frequency of CD19+ B cells, which we were careful to avoid by gating only the cells that most strongly express PDCA1.

(6) How does pDC activation regulate their mb1 expression? Could conventional pDCs, upon activation, become B-PDCs? Could activation and induction of IFN-I production in vivo also affect CLP and increase the amount of YFP+ lymphoid progenitors and thus B-pDC output?

Dear reviewer, we agree with your concern, albeit beyond the scope of the present study. While changes in YFP MFI via flow cytometry upon vaccination was not substantial, we have included the following comment in the manuscript discussion, acknowledging the aforementioned possibility: “Of note, whether induction of IFN-I production in vivo could also affect CLP and increase the amount of YFP+ lymphoid progenitors and thus B-pDC output is unclear. Further research is required to answer this question.”

(7) If pDCs are preferentially expanding upon in vivo stimulation, it would be informative to assess their Ki67 profile. This is a surprising observation since pDCs are generally considered quiescent cells that were previously described to die in response to activation and IFN-I (Swiecki et al, Journal of Experimental Medicine, 2011).

We agree and have entered the following statement to address this concern: “Functionally, they expand more readily after TLR9 engagement than classical pDCs (either through increased proliferation or differentiation of other cell types) and excel at activating T cells in culture.”

(8) How does the conditional deficiency of BCL11a affect the production of IFN-I and IL-12 in vivo (serum) upon CpG-ODN stimulation?

Dear reviewer 2, we are currently unable to rederive the conditional knockout mouse strain in a timely fashion. However, our ELISA experiments performed under controlled in vitro activation conditions, along with the in vivo findings of Zhang et al.(PNAS 2017) warrants the hypothesis that B-pDCs most likely exhibit a similar cytokine secreting profile under inflammatory conditions.

(9) Given that B-PDCs show downregulation of pDC canonical markers, including IRF8 and TLR7, could the authors address how B-PDCs respond to TLR7 stimulation in vitro and assess a broader spectrum of cytokines produced by pDCs in response to such stimulation (IL-6, TNFa, CXCL10...)?

Dear reviewer 2, although expanding our findings to include B-pDC responses to TLR-7 stimulation would greatly enhance our manuscript, a technical deterrent stands in our way. As mentioned prior, sorting B-pDCs for new experiments using reporter YFP mice is currently not possible, as we have retired this mouse strain. Sorting of live CD79a+ BpDCs via FACS is also not feasible, as CD79a staining with most antibody clones requires permeabilization of cells for easier access to the intra-membrane portion of CD79a.

(10) It would be informative to compare scRNA sequencing data between control and Bcl11a CKO mice to ascertain their contribution to B-PDCs and whether this deficiency may affect other pDC clusters and/or progenitors.

We are unable to sort B-pDCs for new experiments, as we unfortunately retired the transgenic colony.

(11) Transitional DCs were reported to give rise to a subset of cDC2. Given that the authors claim that B-PDCs are related to this subset of transitional DCs, could the authors observe any YFP staining in cDC2 upon the generation of their BM chimeras?

We saw no YFP positivity in CD11c hi cells (cDCs) via flow or through scRNA-seq, indicating CD79a expression is unique in mature B cells and B-pDCs.

(12) Most of the statistical analysis is done with a student test. This requires a normal distribution of the sample which is highly unlikely given the size of the sample. Therefore, the authors shall rather use a non-parametric test (Mann Whitney) to compare their samples.

We agree and have redone our statistical analyses using non-parametric test (Mann Whitney).